# Phononic Crystal Made of Silicon Ridges on a Membrane for Liquid Sensing

**DOI:** 10.3390/s23042080

**Published:** 2023-02-13

**Authors:** Abdellatif Gueddida, Victor Zhang, Laurent Carpentier, Jérémy Bonhomme, Bernard Bonello, Yan Pennec, Bahram Djafari-Rouhani

**Affiliations:** 1Institut d’Electronique, Microélectronique et Nanotechnologie, UMR CNRS8520, Université de Lille, 59650 Villeneuve d’Ascq, France; 2Institut des Nanosciences de Paris, Sorbonne Université, UMR CNRS 7588, 75005 Paris, France

**Keywords:** acoustic sensor, phononic crystal, defect mode, liquid viscosity, transmission curve, dispersion curve, membrane

## Abstract

We propose the design of a phononic crystal to sense the acoustic properties of a liquid that is constituted by an array of silicon ridges on a membrane. In contrast to other concepts, the ridges are immersed in the liquid. The introduction of a suitable cavity in the periodic array gives rise to a confined defect mode with high localization in the cavity region and strong solid–liquid interaction, which make it sensitive to the acoustic properties of the liquid. By using a finite element method simulation, we theoretically study the transmission and cavity excitation of an incident flexural wave of the membrane. The observation of the vibrations of this mode can be achieved either outside the area of the phononic crystal or just above the cavity. We discuss the existence of the resonant modes, as well as its quality factor and sensitivity to liquid properties as a function of the geometrical parameters. The performance of the proposed sensor has then been tested to detect the variation in NaI concentration in a NaI–water mixture.

## 1. Introduction

It is well known that periodic phononic crystals (PnCs) exhibit forbidden bands which prevent the propagation of acoustic or elastic waves. This ability finds wide applications in guiding, controlling, and manipulating acoustic and elastic waves [1,2,3,4]. However, localized modes may exist inside the bandgaps when a defect is inserted in a perfect PnC. Such a mode can be used for the purpose of sensing applications, for example, to measure the volumetric properties of a liquid filling a cavity [5,6]. Such a functionality can be developed in a broad range of targeted frequencies, depending on the size of the unit cell in designed PnC systems. The sensing properties may include material parameters, such as mass density, sound velocity, viscosity and their variations with temperature [7], concentration in a mixture solution [8], or the phase transition of the analyte [9]. 

The concept and feasibility of liquid sensors based on 1D multilayered structures and 2D PnCs with holes filled with a liquid were proposed during the 2000s [10]. Other authors have demonstrated high-performance designs by introducing cavities [11,12] slots [13,14], or waveguides [15] in 1D or 2D periodic crystals. A concept based on the ring resonator employing a 2D surface phononic crystal has been proposed to analyze binary gas mixtures facing the ring resonator with high sensitivity [16]. Achieving both the high-quality factor and sensitivity of the characteristic response, i.e., peak or dip in the transmission spectrum, is required in order to reach a sufficient resolution of the successive features.

The recent development of PnC sensors is going toward integration. PnCs made of finite plate thicknesses with hollow pillars filled with a liquid have been both theoretically [17] and experimentally [18] investigated, showing that the resulting localized modes were sensitive to the height of the liquid in the hole, its concentration, or temperature. The application range of pillared PnCs was further extended to include the detection of the mass of nanoparticles [19]. 

Recently, we studied a tubular-type PnC, consisting of a periodic arrangement of cylindrical washers distributed along a tube [20,21,22] in which the liquid flows, for the purpose of sensing the physical properties of the liquid filling the tube. This design has the advantage of avoiding any perturbation of a flowing fluid by any element inside the tube, for instance, a periodic arrangement of its internal section. Finite element simulations allowed us to demonstrate the existence of complete, as well as polarization-dependent, bandgaps, inside which localized modes associated with defects can be found. Their signature appears as peaks or dips in the transmission spectrum. We found the dramatic effect of the liquid viscosity on the transmission features, as well as the significant damping from the polymer used as the material for the tube. 

It should be mentioned that contrary to traditional liquid sensors, in which the inspection of the acoustic properties of the liquid is conducted in the interfacial layer between the solid and the liquid, PnC liquid sensors enable the acoustic waves to penetrate into the whole analyte, thus probing its acoustic properties in the entire volume [6], one can also notice that, thanks to the scalability of frequencies with the structure periodicity, PnC-based sensors can be designed to operate in any required frequency range. They are also interesting for flammable liquids because the electronic components that compose the source and the detector may be easily separated from the liquid being monitored [23]. 

The three main types of transduction techniques extensively studied for mass-sensitive acoustic biosensors are quartz crystal microbalance [24,25,26,27,28], film-bulk acoustic-wave resonators [29,30,31], and surface acoustic waves [32,33,34]. The first two suffer from acoustic radiation in liquids, unless making the device work in shear wave mode. Potential ways to efficiently suppress this kind of radiation loss reside in reducing the useful wave velocity to a value below the sound speed of liquids. Based on this idea, the Lamb wave of lower orders can be exploited, either with very thin plates (a few percent of the wavelength) [35,36,37,38] or with other means, such as using relatively thick interdigitated electrodes [39]. A similar idea was exploited to design a novel Lamb wave resonator for the high-resolution mass-sensitive detection of biomolecules [40]. Using a piezoelectric plate and an interdigitated transducer made of high-aspect-ratio electrodes to excite slow velocity hybrid mode, acoustic radiation in water was largely suppressed. The authors improved the overall sensing resolution by achieving high-quality factor resonance in water, increasing the mass sensitivity and reducing the noise frequency. The involved elastic resonance arose from a combination of the fundamental Lamb mode and the flexural mode of the electrodes. 

In this paper, we focus on a PnC structure that is similar in shape but conceptually different, consisting of a silicon membrane supporting, on one side, high-aspect-ratio ridges immersed into the liquid volume. Indeed, we are interested in the design of a cavity that can support a confined mode with sufficient solid–liquid interaction and yield a very good quality of detection. This architecture permits the probing of the acoustic properties within the maximum liquid volume, while producing significant vibrations in the solid membrane. Therefore, detection can be conducted both at the exit of the membrane and at the top of the cavity, for instance, by a laser vibrometry technique. We analyze the effect of mass density and sound velocity variations in the liquid on the characteristics of acoustic transmission through the PnC. For a given PnC design that gives rise to a defect mode within the bandgap, we demonstrate that the quality factors of the frequency response can be improved by increasing the number of ridges to reach the *Q*-factor of the order of hundreds with about a tenth of ridges. We also study the effect of liquid viscosity on the characteristics of the sensor.

The outline of the paper is as follows. In Section 2, we study the frequency characteristics of the PnC and the solid–fluid interaction by means of dispersion and transmission curves. They show the existence of bandgaps and are helpful in understanding their physical origin. In Section 3, a particular type of defect is introduced into the PnC, and the characteristics of the associated defect modes (in particular, the *Q*-factor) are investigated by varying the defect parameter and the number of ridges. Section 4 deals with the sensing performance of the designed PnC with lossless liquids, while Section 5 considers the effects of liquid viscosity on the transmission responses. In Section 6, the sensor is tested to detect NaI concentration in the NaI–water mixture. Finally, some conclusions are drawn in Section 7.

## 2. Dispersion Curves and Transmission Spectra

The PnC that we study (see Figure 1) consists of a silicon membrane (thickness *h*_1_ = 100 µm) and a periodic arrangement of silicon ridges of length *h_p_* = 800 µm and thickness *d* = 100 µm with the period *a* = 628 µm. The ridges are immersed in the liquid to be sensed, which is therefore in direct contact with the membrane. The elastic properties used in the calculation for the silicon membrane and ridges are *C*_11_ = 165.7 GPa, *C*_12_ = 63.9 GPa, *C*_44_ = 79.9 GPa, and *ρ* = 2330 kg/m^3^, whereas the liquid medium (water) is defined by its sound velocity *C_lf_* = 1490 m/s and its mass density *ρ_f_* = 1000 kg/m^3^. This geometry is preferable compared to the case where the ridges are on the air side because the solid–liquid interaction is less efficient in the latter case. The liquid is assumed to be very thick, so in the simulation Plane Wave Radiation, conditions are used at the bottom of the structure, as well as the Perfectly Matched Layer (*PML*), at both vertical sides of the structures to allow the waves travelling towards the structure’s boundaries to escape the simulation area. Due to the infinite length of the ridges in the lateral (*z*) direction and the weak coupling between the liquid and the shear components of the displacement field associated with the (*z*) direction, the simulation was conducted in two-dimensional (2D) geometry. Here, we focus on the case of excitation by a transverse force *F_y_*, which gives the most promising results, and the PnC does, in general, constitute 12 to 18 ridges. It should be noted that the experimental implementation can be conducted with conventional techniques, such as the use of piezoelectric transducers to excite and detect transverse waves. The harmonic force Fy is applied by a transducer on top of the membrane operating in piston mode. The measurement of the transmission coefficient is conducted thanks to a second transducer whose bandwidth is calibrated and placed on top or downstream of the phononic crystal [41]. When a non-contact approach is required, a laser-ultrasound technique is more relevant. In short, it consists of exciting the membrane into vibration by an ultra-short light pulse (typically a few ps to a few ns) and detecting the out-of-plane motion of the surface by an interferometer [42], or by an optical heterodyne technique [43].

The numerical simulations are performed by COMSOL Multiphysics® (Stockholm, Sweden), using the Solid Mechanics module and the Pressure Acoustic module to describe the elastic wave propagation in the solid part of the structure and the pressure variation in the liquid part, respectively [44]. The dispersion curves, corresponding to a perfect periodic array of ridges, are first calculated by considering the elementary unit cell shown in Figure 2a. Periodic boundary conditions are applied at two opposite edges of the unit cell, perpendicular to the propagation direction (*x* direction), whereas plane wave radiation conditions and free boundary conditions are used at the bottom limit of the unit cell and at the air/solid interface of the membrane, respectively (Figure 2a). As shown in Figure 2b, the modes in the dispersion curves can be separated into two sets depending on their frequencies with respect to the sound velocity of the liquid (indicated by the black line in Figure 2b). Indeed, the modes located below this sound line cannot propagate in the liquid and are localized in the solid part with weak acoustic radiation in the liquid, while the modes located above the sound line propagate in both the solid and the liquid. The colors on the branches correspond to the ratio between the *U_y_* component of the displacement field in the solid part and the total displacement field of the whole unit cell. Figure 2c shows the transmission curves obtained for the vertical excitation (*F_y_* = 1 N/m) of the membrane. The obtained transmission curve was calculated by considering a finite PnC made up of twelve ridges. Elastic wave generation is carried out by applying a force *F_y_* loaded at the entry of the PnC, then the transmission curve is then obtained by probing the displacement field at the outlet of the PnC and normalized to the displacement field evaluated for the membrane without the PnC. One can remark the presence of two bandgaps indicated by the gray area on the graphs, located at around 550 and 1150 kHz. As highlighted by the colors on the branches of the dispersion curves, the obtained transmission curve corresponds to the excitation of the branches with relatively strong vertical components of the displacement field. In what follows, we focus our study on the first bandgap, in which the acoustic radiation of the excited flexural (A_0_ mode) mode of the silicon membrane in the liquid is relatively weak. 

## 3. Design of a Cavity and Defect Modes

We consider a transverse excitation (*F_y_* = 1 N/m) to launch the incident wave and focus on the frequency range of the first bandgap at around 550 kHz where we shall demonstrate the existence of a confined mode that is sensitive to the properties of the liquid surrounding the ridges. The considered cavity is obtained by varying the period of the PnC at the middle. A parameter *a*_0_ is defined by the ratio *a*_0_ = *a_d_/a*, where *a_d_* is the space between the two ridges in the middle of the PnC (Figure 3). A value of *a*_0_ ≥ 1 or *a*_0_ ≤ 1 implies that the two parts of the PnC are moved away or brought closer to each other.

We chose the quality factor of the relevant peak and its position in the transmission spectrum as the criteria for the choice of the parameter *a*_0_. Figure 4 displays the transmission curves of the PnC cavity calculated for different values of the parameter *a*_0_. The transmission curves are separated into two graphs according to the value of the parameter *a_d_* of the cavity compared to the lattice parameter *a* of the perfect PnC. The left and right panels (Figure 4a,b) gather the transmission curves of the cavity that meet the conditions *a*_0_ ≥ 1 and *a*_0_ < 1, respectively. The simulation results show that a defect mode appears in the bandgap around *f*~(400–800) kHz for some values of the parameter *a*_0_, as indicated by the transmission peak, and this peak shifts to lower or higher frequencies as the two parts of the PnC move away (*a*_0_ ≥ 1, Figure 4a) or becomes closer (*a*_0_ ≤ 1, Figure 4b), respectively. Note that due to the finite size of the PnC, the limits of the bandgap change slightly with *a*_0_, as does the dip immediately next to the peak at *f* = 450 kHz. With cavity parameter *a_d_* greater than the lattice parameter of the perfect PnC (*a*_0_ ≥ 1), the associated transmission curves give rise to peaks with a poor quality factor. The only one with a relatively high *Q*-factor (*a*_0_ = 1.2) is located near the bandgap edge, which is not suitable for realizing the sensing functionality. In contrast, with a cavity parameter *a_d_* less than the lattice parameter of the perfect PnC (*a*_0_ ≤ 1), the obtained transmission curves provide peaks with a relatively high *Q*-factor and for some *a*_0_ values, and it is possible to obtain peaks in the middle of the phononic bandgap. Consequently, the *a*_0_ parameter value that complies the aforementioned criteria is *a*_0_ = 0.6. In order to understand the physical origin of the associated peaks, in Figure 4c, we present both the displacement field and the absolute acoustic pressure of the cavity modes associated with the two values of the parameter *a*_0_: 0.6 and 1.5. One can observe that for *a*_0_ ≤ 1, the cavity mode corresponds to the strong localization of the displacement field in both ridges constituting the cavity and in the liquid at the bottom of the ridges, thus resulting in the weak radiation of the confined mode in the liquid. When the space between the two ridges exceeds the value of the lattice parameter *a*, e.g., *a*_0_ = 1.5, the acoustic radiation of the vibrating membrane increases in the liquid, which results in a decrease in the *Q*-factor of the peaks in the transmission curves (Figure 4a).

The highest *Q*-factor shown in Figure 4b is about 70 for *a*_0_ equal to 0.6. However, this value can be much improved by increasing the number of the ridges in the PnC [21]. Figure 5 presents the transmission spectrum of the relevant peak calculated for the PnC made up of three sets of ridges, i.e., 12, 14, and 18 ridges. One can remark that the quality factor increases from 70 with 12 ridges to 500 with 18 ridges. This increase is beneficial to both increase the figure of merit of the sensor and limit the loss of the quality factor when the viscosity is introduced into the simulation (Section 5). 

Figure 6 shows the displacement field recorded along the air/membrane interface at the resonance frequency of the cavity. It shows a strong vibration of the membrane in the region of the cavity, which is about 15 times higher in comparison with the detector region. This gives an additional and alternative way to measure the frequency response of the structure, particularly in the case of relatively highly viscous liquids where the transmission peak is highly altered.

## 4. Sensitivity of the Sensor to the Acoustic Properties of the Liquid

We have seen that the best detection of the transmission peak is obtained with a shrinked cavity of *a*_0_ = 0.6, where a *Q*-factor of around 500 can be achieved by using 18 ridges in the PnC. In addition to the *Q*-factor, the efficiency of the acoustic sensor can be evaluated from the change in the transmission peak when changing the parameters of the liquid, such as the sound velocity or the mass density. This is illustrated in Figure 7a, in which we start with water as the reference liquid and change either its velocity or density by 10%. It can be observed that the proposed sensor is mostly sensitive to the mass density *ρ_f_* (with *Δf/f*~1.17%), while a variation in sound velocity does not significantly affect the resonance peak (*Δf/f*~−0.175%). Of course, in a specific experiment in which the concentration, temperature, or phase of an analyte is the varying parameter, both the sound velocity and mass density are subject to variations and the detected change in the resonance frequency is a combination of both effects. This is illustrated in Section 6 in which we change the concentration of NaI in a NaI–water mixture.

From these results, one can also derive two characteristic parameters of a sensor, namely the sensitivity (*S*) and the figure of merit (*FoM*), which are defined as follows:(1)$$S_{Clf}=\frac{\Delta f_{r}}{\Delta Clf\left(\%\right)}\;(\mathrm{Hz}/\%)\;\mathrm{or}\;S_{\rho f}=\frac{\Delta f_{r}}{\Delta \rho f\left(\%\right)}\;(\mathrm{Hz}/\%)$$
(2)$$FoM_{Clf}=\frac{Q.S_{Clf}}{f_{r}}\;(\%{)}^{-1}\;\mathrm{or}\;FoM_{\rho f}=\frac{Q.S_{\rho f}}{f_{r}}\;(\%{)}^{-1}$$

Here, the sensitivity (S) represents the ratio of the frequency shift $\Delta f_{r}$ of the transmission peak (cavity mode) with respect to the change in either the velocity or the mass density expressed in % of its initial value. Then, the inverse of the figure of merit (*FOM*) has the ability to define the smallest variation in the acoustic property that can be detected, namely when the frequency shift, due to the change in the acoustic property, exceeds the width of the peak. Figure 7b shows that the frequency peak changes almost linearly with both *ρ_f_* and *C_lf_*, and the slopes obtained from the linear regression of the frequency peak variation give the sensitivities $S_{Clf}=90$ Hz/% (*R^2^* = 0.993) and $S_{\rho f}=600$ Hz/% (*R^2^* = 0.999), respectively. The associated figures of merit are $FoM_{Clf}=0.087$ (%)^−1^ and $FoM_{\rho f}=0.5$ (%)^−1^. Thus, the minimum detectable values of *ρ_f_* and *C_lf_* are evaluated to be 20 kg/m^3^ and 200 m/s, respectively. Again, this confirms the higher ability of the design to detect a variation in the mass density than in the acoustic velocity. In the Appendix A, we briefly present a similar design with stubs immersed in water but without a PnC that is more suitable for the detection of sound velocity.

## 5. Effects of Liquid Viscosity on Frequency Response

In most liquid sensors, the viscosity is a very limiting factor because it strongly decreases the *Q*-factors of the peaks. More particularly, in most PnC crystal sensors, this effect has often been omitted or not been taken deeply into consideration by neglecting the major effect due to the dynamic viscosity, which is at the origin of dissipation in the boundary layers. To incorporate this effect into our study, we used a more rigorous liquid model, called the Thermoviscous Acoustic module of COMSOL Multiphysics software, which takes into account both dynamic (*µ*) and bulk (*µ_B_*) viscosities [44]. We recalculated the transmission curves in the frequency range of 505 to 530 kHz, which covers the involved defect mode. With the known values for water viscosity under normal atmospheric conditions, i.e., *µ* = 0.89 and *µ_B_* = 2.3 (mPa.s), the results are shown in Figure 8. It shows a dramatic effect of the viscosity on the peak amplitudes by a factor of about 4, hence a reduction in the associated quality factor from 500 to 285. Accordingly, this affects the *FOMs*, while the sensitivities $S_{Clf}$ and $S_{\rho f}$ remain unchanged. In addition, it should be noticed that the amplitude of the transmission peak is also influenced by the variation in both velocity and mass density; however, this effect remains weak regarding that of the viscosity, particularly for small variations in velocity or mass density. From the above analysis, one can also conclude that the amplitude of the peak provides at least a qualitative tool for estimating liquid viscosity. It should be noted that the amplitude of the resonance or its *Q*-factor may be further affected if the membrane is made of a dissipative solid material, such as a polymer [21]. In the present work, the choice of silicon with low acoustic dissipation avoids this drawback.

## 6. Sensing of NaI Concentration in NaI–Water Mixtures

In the previous sections, we investigated the sensitivity (*S*) and figure of merit (*FoM*) of the proposed phononic crystal sensor with respect to hypothetical variations in the sound velocity and the mass density of the liquid. In this section, we apply our design to a practical problem of interest, namely testing the ability of the sensor to detect variations in NaI concentration in a NaI–water mixture. Figure 9a gives both the experimental sound velocity *C_lf_* (red line) and the mass density *ρ_f_* (blue line) of the NaI–water mixture for different concentrations of NaI from 0 to 45%, taken from reference [45]. The graphs show an almost linear relationship between the mass density and concentration of NaI with an increase of about 50% when the concentration reaches 45%, while the sound velocity presents a minimum at a concentration of 6% and displays a variation of less than 3% over the whole range of concentration. Similarly, Figure 6b shows the variations in the dynamic and bulk viscosities of the NaI solutions as a function of the concentration [45]. The former, which has the main effect on the quality factor of the resonances, is almost a constant, while the latter increases with the concentration. With these parameters, the transmission peak was calculated for different concentrations of NaI (0 to 45%). As shown in Figure 9c, despite the viscosity effect, the obtained peaks are quite distinguishable for different measured NaI concentrations. One can note a redshift in the resonance frequency with an increase in NaI concentration, accompanied by an increase in its amplitude in the transmission spectrum. The redshift frequency is mainly due to the increase in mass density for higher concentrations. Of course, there is also an effect due to the change in sound velocity, but this effect remains relatively small and does not change the observed trend, despite the presence of a minimum in the velocity (Figure 9a). On the other hand, the variation in the amplitude of the transmission is related to the position of the resonance frequency in the middle of the bandgap and the associated *Q*-factor. This explains that the amplitude and *Q*-factor of the resonances increase despite an increase in liquid viscosity. Figure 9d displays the frequency peak as a function of the NaI concentration. It shows that this frequency changes almost linearly, and the slope of the linear regression of the associated curve gives the sensitivity *S* = 566 Hz/% (*R^2^* = 0.997). 

## 7. Conclusions

We investigated the transmission properties of a PnC based on periodic ridges arranged along a membrane, both made of silicon. Numerical results were obtained using the COMSOL Multiphysics simulation for the test structure. We demonstrated that the mixed solid/liquid system can present bandgaps for flexural Lamb waves. The introduction of a defect into the PnC structure, by modifying the separation distance of the two parts of the same PnC, leads to the occurrence of a peak inside the bandgap in the transmission spectrum. The peak frequency position has been shown to be mainly sensitive to the mass density and much less to the sound velocity of the liquid in contact with the membrane and immerging the ridges. The transmission peak is due to the cavity mode, whose frequency position depends on the liquid properties in addition to the geometric parameters of the PnC. We highlighted the significant effect of viscosity on the amplitude of the transmission peak. Such an effect can be partially avoided by probing the elastic vibration in the cavity region instead of the detection region after the PnC with the same experimental method. Our sensor demonstrated its ability to detect changes in NaI concentration in a NaI–water mixture. The proposed device would be a useful platform for sensing liquid properties in various fields and at different scales, from microfluidic to medical or civil engineering applications. 

## Figures and Tables

**Figure 1 sensors-23-02080-f001:**
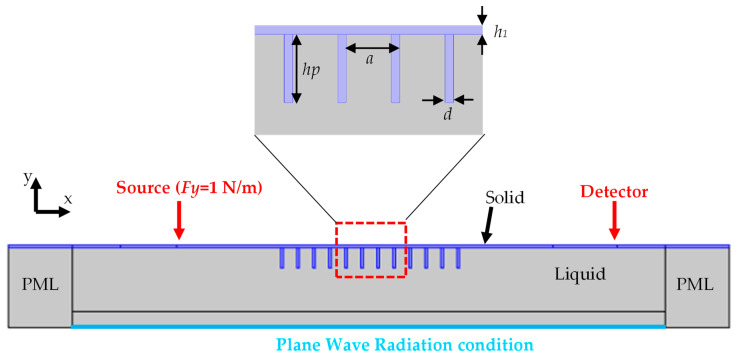
Schematic model used to study the transmission of elastic waves along a membrane and through a PnC of twelve periodic ridges excited by a transverse force *F_y_* (1 N/m). The membrane has a thickness *h*_1_ = 100 µm, the ridges are defined by their length *h_p_* = 800 µm and their thickness *d* = 100 µm, and the period of the PnC is *a* = 628 µm.

**Figure 2 sensors-23-02080-f002:**
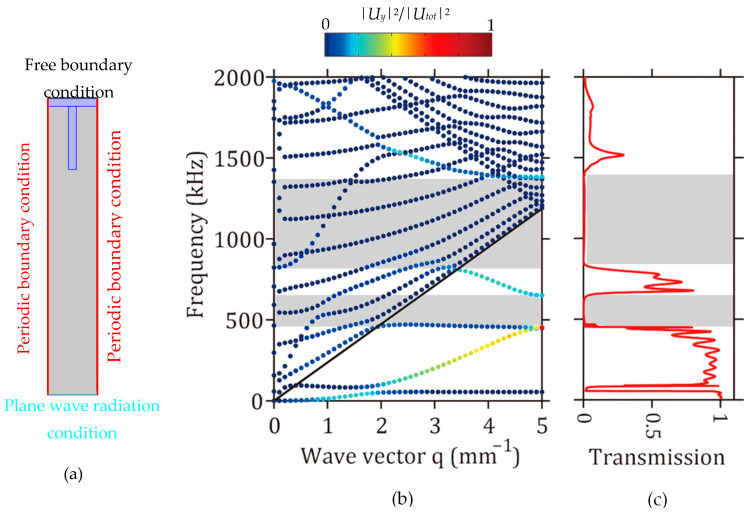
(**a**) Unit cell used for the calculation of the dispersion curves, (**b**) dispersion and (**c**) transmission curves considering 12 ridges obtained with parameters *a* = 628 µm, *d* = 100 µm, *h_p_* = 800 µm. The black line represents the sound velocity in water, the color on the branches corresponds to the ratio between the *Uy* component in the solid part, and the total displacement field in the whole structure. The gray areas indicate the bandgap regions.

**Figure 3 sensors-23-02080-f003:**
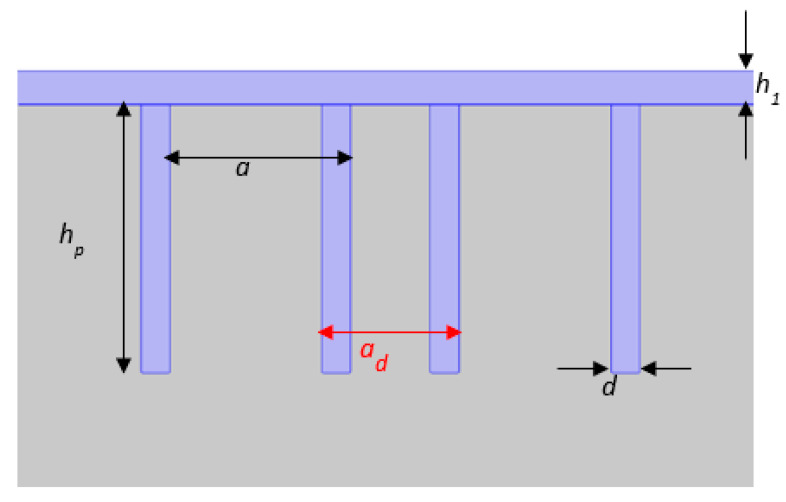
Zoom around the cavity created in the middle of the PnC. The cavity is defined by the geometrical parameter *a_d_* indicated by the red arrow.

**Figure 4 sensors-23-02080-f004:**
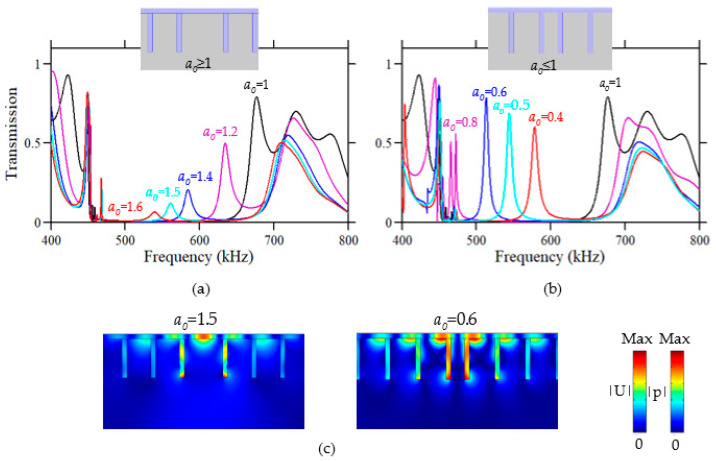
Frequency responses of the displacement excited by a transverse force and detected on the opposite side of a PnC of 12 ridges with defect (**a**) *a*_0_ ≥ 1 and (**b**) *a*_0_ ≤ 1, showing the variation in the peaks with a value of *a*_0_. The insets show a schematic view of the PnCs with a central defect. (**c**) Displacement field (solid) and absolute acoustic pressure (liquid) maps obtained with the parameters *a*_0_ = 1.5 and *a*_0_ = 0.6.

**Figure 5 sensors-23-02080-f005:**
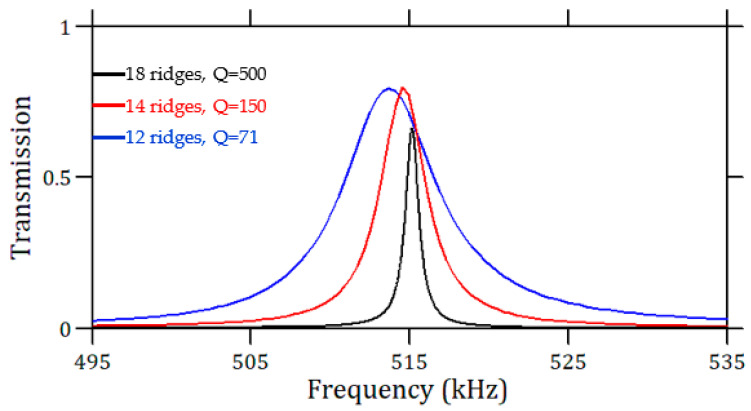
Comparison of transmission peak calculated for the PnC made of 12, 14, and 18 ridges. The inserted cavity corresponds to *a*_0_ = 0.6.

**Figure 6 sensors-23-02080-f006:**
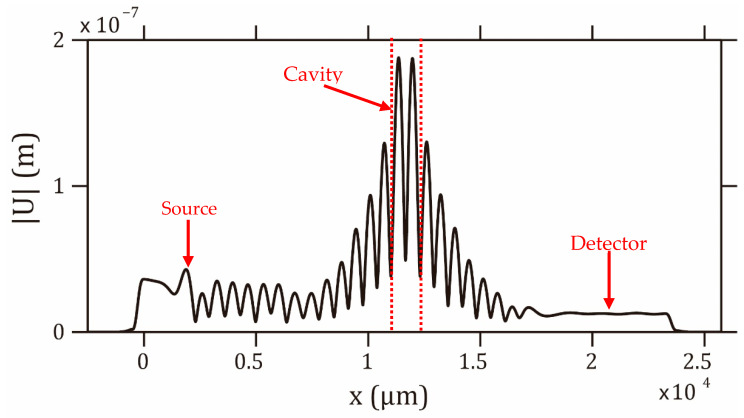
Displacement field recorded on the air/membrane interface at the cavity mode frequency. The PnC is made of 18 ridges.

**Figure 7 sensors-23-02080-f007:**
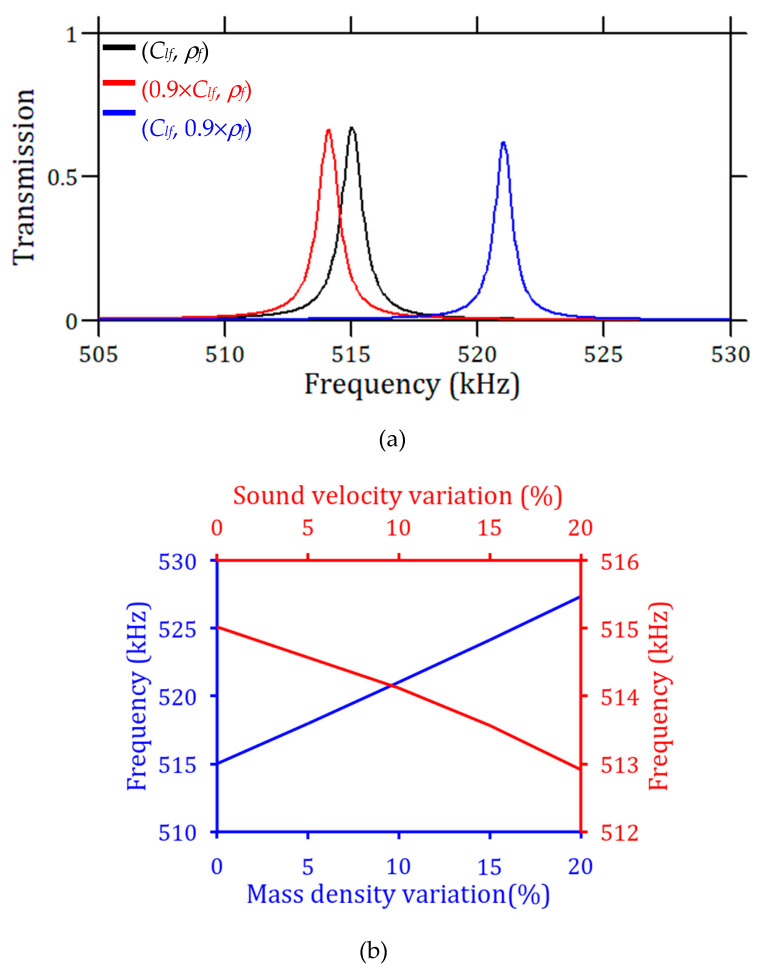
(**a**) Transmission peaks with three non-dissipative liquids of different acoustic properties. The PnC is defined by *h_p_* = 800µm, 18 ridges, and *a*_0_ = 0.6 The black curve corresponds to the reference liquid, i.e., water (*C_lf_* = 1490 m/s and *ρ_f_* = 1000 kg/m^3^). The red and blue curves correspond to a change in acoustic velocity by 10% (*C_lf_* = 1341 m/s, *ρ_f_* = 1000 kg/m^3^) or mass density by 10% (*C_lf_* = 1490 m/s, *ρ_f_* = 900 kg/m^3^), respectively. (**b**) Transmission peak frequency as a function of the percentage change in mass density *ρ_f_* (blue curve) and sound velocity *C_lf_* (red curve).

**Figure 8 sensors-23-02080-f008:**
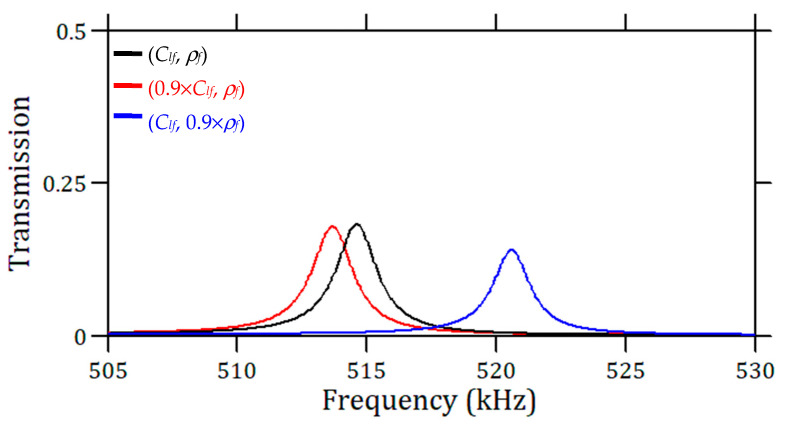
Transmission peaks with three dissipative liquids of different acoustic properties. The parameters of the PnC are *h_p_* = 800 µm, 18 ridges, and *a*_0_ = 0.6. The black curve corresponds to the reference liquid, i.e., water (*C_lf_* = 1490 m/s and *ρf* = 1000 kg/m^3^). The red and the blue curves correspond to (*C_lf_* = 1341 m/s, *ρ_f_* = 1000 kg/m^3^) and (*C_lf_* = 1490 m/s, *ρ_f_* = 900 kg/m^3^), respectively. The viscosity of liquids is taken for all the three cases as *μ* = 0.89 mPa.s and *μ_B_* = 2.3 mPa s.

**Figure 9 sensors-23-02080-f009:**
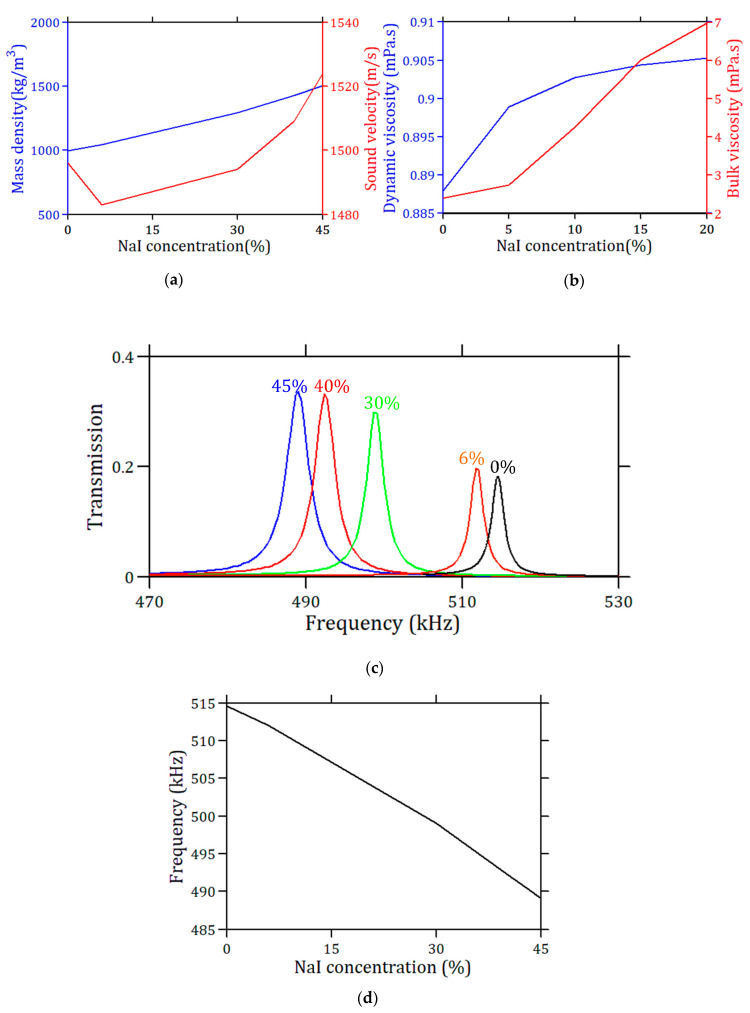
(**a**) Sound velocity and mass density of NaI–water mixture for different NaI concentrations [45]. (**b**) Dynamic and bulk viscosities of NaI–water mixture for different NaI concentrations. (**c**) Transmission peak of the phononic cavity as a function of the NaI concentration. (**d**) Frequency of the peaks as a function of the associated NaI concentration.

## Data Availability

The data that support the findings of this study are available from the corresponding author upon reasonable request.

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
