# Peer review of "Phononic Crystal Made of Silicon Ridges on a Membrane for Liquid Sensing"

_sensors, 2023, doi:10.3390/s23042080_

Round 1

Reviewer 1 Report

In this manuscript, the authors proposed the design of a phononic crystal for sensing the acoustic properties of a liquid based on silicon ridges on a membrane. This topic is interesting and promising. However, there are some questions exist in the manuscript; it should be revised considering the suggestions as following:

1.    In the introduction, the advantage of the new liquid sensor structure compared with the traditional liquid sensors such as Love sensor should be stated more clearly.

2.    In the part 2, the model for the dispersion calculation should be added whether it is a periodic structure model. For the transmission calculation, how to guarantee the excitation and detection of the wave mode is expected.

3.    Additionally, in the Figure 2(a), which range of the wave vector is chosen and the reason for this should be given.

4.    From the Figure 5, the number of the ridges not only influences the Q but also influence the amplitude of the transmission. How to choose the number of the ridges should be give more detail information.

5.    In the Figure 6, the horizontal coordinate maybe have some mistake, please confirm this information.

6.    In the part 5, the transmission of this structure may be influenced by both the velocity and the viscosity. Can the authors give more information for analysis for the decoupling for different variations?

7.    In the part 6, what range of the concentrations of NaI can this device measure? Is it just 0% to 45%? Could the authors give the reason for this? In this section, it may be more significant if the authors give the comparison of the theoretical and experimental results.

Reviewer 2 Report

The authors numerically proposed a phononic crystal sensor to detect variations in mass density and sound velocity of a liquid solution. The variations of the acoustic parameters caused by the variation of solution can be quantitatively related to the shift and Q factor of a resonance peak in a defect phononic crystal. They also define sensitivity and figure of merit as the characteristic parameters of the sensor. The manuscript is well rewritten. There are no evident mistakes as I can tell. Overall, I think it is suitable for the publication in Sensors as this work could meet many readers’ interests. I recommend the publication until the following comments are addressed.

1.       The authors use a transverse excitation (y-direction) to excite the PnC. What would happen if the excitation was switched to x-direction? Would the performance remain nearly the same?

2.       What could be used as the source to excite propagating waves in this system?

3.       I wonder if the silicon membrane would possess damping effect. What would happen if the silicon membrane had some damping? Can the authors comment on that?

Minor revision:

At line 263, I think “Fig. 7” should be replaced with “Fig. 8”.

Reviewer 3 Report

Authors introduce a photonics crystal with ridges for detecting characteristics of liquids. By introducing defect (cavity) modes in the photonic crystal structure they generate a peak inside the band gap that increases the sensitivity of the sensor to mass density of the liquid. Through the experiments they have demonstrated the capability to detect NaI concentration in NaI water mixture.
The manuscript is very well-written, and all relevant research work has been cited. I recommend this MS to be published with minor (editorial) revisions.

Author Response

We thank the reviewers for their helpful comments and recommendations that allow us to improve the quality of our manuscript.

Round 2

Reviewer 1 Report

The authors answer well for the initial comments.